# Preparation of Wide-Domain pH Color-Changing Nanocapsules and Application in Hydrogel Fibers

**DOI:** 10.3390/ma15248787

**Published:** 2022-12-09

**Authors:** Xuemei Hou, Huijing Zhao, Ke-Qin Zhang, Kai Meng

**Affiliations:** 1College of Textile and Clothing Engineering, Soochow University, No. 178 Ganjiang Road, Suzhou 215006, China; 2National Engineering Laboratory for Modern Silk (Suzhou), No. 199 Ren’ai Road, Industrial Park, Suzhou 215123, China

**Keywords:** pH color-changing, nanocapsules, flash nanoprecipitation, pH indicator

## Abstract

In recent years, there has been an increase in demand for pH color-changing materials. These materials can visually communicate signals to people by connecting pH changes with color information. Embedding pH indicators into fibers to create flexible color-changing materials is an effective way to develop daily wearable products. For the stability of the indicator and the indirect contact of the indicator with the human body, it is usually necessary to encapsulate it in capsules. In this study, different pH indicators (Thymol Blue-TB, Bromocresol Green-BCG, and Bromocresol Purple-BCP) were mixed into a wide-domain pH color-changing indicator and encapsulated with ethyl cellulose (EC) by the flash nanoprecipitation (FNP) method using a new-type droplet-shaped confined impinging jet mixer. The effects of flow rate, core-to-wall ratio, and mixed solution concentration on the formation of the nanocapsules were investigated. In addition, the morphology, particle size, size distribution, dispersion stability, and encapsulation efficiency were systematically studied. At a core-to-wall ratio of 1:2, a mixed solution with a concentration of 6 mg/mL and a feed flow rate of 40 mL/min produced nanocapsules with an average particle size of 141.83 ± 0.98 nm and a PDI of 0.125 ± 0.01. Furthermore, a zeta potential with a range of −31.83 ± 0.23 mV and an encapsulation efficiency of 75.20 ± 1.72% were observed at 1:2 core-to-wall ratios. It was concluded that the color of the nanocapsules continuously changed from yellow to green and green to blue when the pH range was increased from 3 to 10. The color-changing nanocapsules were then embedded into sodium alginate hydrogel fibers, resulting in the same color-changing trend (pH 3–10) as that obtained for the nanocapsules. This study can be useful for the pH monitoring of various body fluids, such as wound exudate, urine, and sweat.

## 1. Introduction

The pH color-changing materials change their color according to the pH value and can lead to sensitive, controlled, and visible environmental response signals in a nondestructive manner. Developing smart color-changing materials with good flexibility and mechanical stability is considered an important new smart textile material with a wide range of applications in different areas. Generally, pH color-changing materials are mainly detected by combining pH color-changing substances with novel chemical substrates through doping, adsorption, chemical bonding, and coating [1]. Schueren [2] studied color-changing textile materials with pH sensitivity, which were based on the dyeing of conventional textiles with standard water-soluble pH indicator dyes. The pH indicator dyes were successfully applied to textile materials, leading to their potential role in textile pH sensors. Liu [3] modified the pH indicator phenol red with methacrylate, copolymerized it into the double net hydrogel matrix, and synthesized a series of pH-indicating colorimetric alginate/P (AAm-MAPR) hydrogels, leading to their application in wound dressing. The authors of [4] prepared fluorophore-NP nanoprobes by linking pH-sensitive fluorescent dyes of different pKa values to gold/silver metal nanoparticles, enabling linear detection of pH values from 5 to 11.

Compared to natural pH indicators, chemical reagents are more sensitive to discoloration, easier to access, and lower in cost. However, the direct use of a single chemical pH indicator may lead to poor stability, contact toxicity, and a narrow indication domain [2,5,6,7], which can limit its application [8]. Therefore, it is necessary to create capsules by mixing different indicators together [9]. The nanocapsules isolate the core material from the external environment, avoiding loss and reaction of the core material due to changes in the external environment. This situation helps to increase nanocapsule stability while also controlling particle size at the nanoscale, which can lead to improvements in material utilization efficiency.

The flash nanoprecipitation (FNP) method is a novel processing technique used for the preparation of nanoparticles that is based on the kinetically controlled principle of turbulent fluid mixing in chemical engineering [10]. Generally, non-water-soluble solutes and polymers are dissolved in organic solvents and mixed vigorously with the counter-solvent in a closed chamber at a specific speed. The solute is supersaturated to form hydrophobic cores and is wrapped to form nanoparticles [11,12]. The confined impinging jet (CIJ) mixer has been widely used in experimental studies related to the FNP technique, where mixing speeds of milliseconds can be achieved [13,14]. In this study, a new type of droplet-shaped mixer was utilized to prepare nanoparticles, which can significantly reduce the particle size while increasing the solubility and utilization efficiency of the nanoparticles.

Hydrogel fibers have the functional properties of high water content, high elasticity, and stimulus response to materials. These properties of hydrogel fibers contribute to materials with a high specific surface area and easy weaving fibrous properties [15]. In addition to these properties, functionalized hydrogel fibers with pH color-changing properties can provide a broader range of applications.

Considering all the facts together, this study addresses the mixing of three pH indicators (Thymol Blue, Bromocresol Green, and Bromocresol Purple) to create a new pH indicator, which can broaden the pH color-changing domain. Then, a wide range of pH color-changing nanocapsules were prepared by a droplet-shaped CIJ mixer with the mixed indicator acting as the core material and ethyl cellulose as the wall material. In the pH color-changing material field, these nanocapsules are firmly bonded with substrates in various forms to develop pH color-changing-related products. Furthermore, the potential applications of the nanocapsules in sodium alginate hydrogel fibers were also explored in this study. The nanocapsules created in this study can be useful for monitoring body fluids, such as sweat, wound exudate, and urine.

## 2. Materials and Methods

### 2.1. Materials

Ethyl cellulose (EC, viscosity: 18–22 mPa·s) and polyvinyl alcohol 1788 (PVA, alcoholic solubility: 87.0–89.0%) were purchased from Aladdin (Shanghai, China). Thymol Blue (TB) and Bromocresol Green (BCG) were obtained from Macklin (Shanghai, China). Bromocresol Purple (BCP) was purchased from Yonghua Chemical Co., Ltd. (Suzhou, China). Ethanol (AR) was obtained from Titanchem (Shanghai, China). pH buffer solution (pH = 3–10) was purchased from Jiangbiao Testing Technology Co., Ltd. (Yichun, China). Deionized water (from the Milli-Q water purification system) was used in all experiments. All chemicals were used without any further purification.

### 2.2. Preparation of Nanocapsules

The pH indicators TB, BCG, and BCP were mixed in a 1:1:1 ratio. TB-BCG-BCP (0.2 g) and EC (0.4 g) were dissolved in 100 mL of ethanol solution (70%, *v*/*v*) and stirred magnetically at 600 rpm for 3 h at room temperature until the indicators were completely dissolved in the solution. The resulting solution was used for the preparation of a mixed solution with a total concentration of 6 mg/mL and a core-to-wall ratio of 1:2. In addition, mixed solutions with mass concentrations of 8 and 10 mg/mL and core-to-wall ratios of 1:1 and 1:3 were separately prepared. A PVA solution with a mass concentration of 0.1 mg/mL was prepared by mixing 0.01 g PVA in 100 mL deionized water at room temperature at 600 revolutions per minute (RPM) for 1 h until the PVA was fully dissolved.

The process of preparing nanocapsules by using the FNP method is shown in Figure 1. Two syringe pumps were used to inject a mixture of TB-BCG-BCP, EC (as the positive solvent), and aqueous PVA solution (the antisolvent) at the same flow rate into the two inlets of the new-type droplet-shaped CIJ mixer. The two fluids then collided in the droplet-shaped CIJ mixer at different flow rates (20, 40, and 60 mL/min). The collisionally mixed liquid stream flows from the mixer’s outlet to a beaker consisting of deionized water to form the nanocapsule suspension. Afterward, the residual ethanol in the nanocapsule suspension was removed using a rotary evaporator.

### 2.3. Characterization Method

#### 2.3.1. Scanning Electron Microscopy

The morphology of the nanoparticles was examined by SEM, model S-4800 (HITACHI, Tokyo, Japan). The suspension of nanocapsules was dropped on a silicon wafer. When the silicon wafer was dry, the nanocapsules were subjected to conductive adhesive and sprayed with gold for 90 s. The scanning electron microscope voltage was set at 3.0 kV.

#### 2.3.2. Fourier Transform Infrared Spectroscopy (FTIR)

Pure TB-BCG-BCP, EC, and EC/TB-BCG-BCP mixtures and discolored nanocapsules (NCs) were tested separately using FTIR spectroscopy, on the NICOLET5700 iS5 (Thermo Electron Corporation, Waltham, MA, USA). Specimens were prepared by the potassium bromide press method and scanned within the wavelength range of 500–4000 cm^−1^.

#### 2.3.3. Particle Size and Distribution Test

The prepared nanocapsule suspension was centrifuged by the centrifuge H3-18KR (Hunan Kecheng Instrument Equipment Co., Ltd., Changsha, China), and the precipitates were taken for dilution. The particle size, dispersity index (PDI), and particle size distribution of the nanocapsules were measured using the Nano ZS90 Malvern laser particle size analyzer (Malvern, Malvern, UK). The measurements were repeated three times, and the results were averaged.

#### 2.3.4. Zeta Potential Test

The zeta potential of the nanocapsule suspension was tested using the Nano ZS90 Malvern Zeta Potentiometer (Malvern, Malvern, UK). Each sample was tested in triplicate.

#### 2.3.5. Encapsulation Efficiency (EE%)

TB-BCG-BCP solutions were prepared at different mass concentrations and subjected to UV full-spectrum scanning. The wavelengths were set from 350 to 600 nm to find the maximum absorption value in the visible range.

The absorbance of the centrifuged TB-BCG-BCP supernatant was measured at the maximum absorption wavelength using the UV-Vis spectrophotometer, Specord S600 (Analytik Jena AG, Jena, Germany). The mass concentration of the supernatant was calculated according to the standard curve equation. The encapsulation efficiency was measured according to Formula (1):(1)$$\mathrm{EE}(\%)=(1\;-\;\frac{m}{M})\;\times \;100,$$
where M represents the total amount of TB-BCG-BCP added, and m denotes the amount of TB-BCG-BCP in the supernatant after centrifugation.

#### 2.3.6. pH Color-Changing Responsiveness Test

The method for quantifying color information is based on the HSV color model, where the HSV color space describes the specific color through the three components of hue (H), saturation (S), and value (V) [16,17]. In this study, the color component H-value is separated, thus quantifying the color change information to explain the relationship between the pH value and the color.

The color changes of the nanocapsules in different pH environments were photographed by a smartphone camera (64 MP) in natural light. The color information was then quantified by identifying and extracting the H-value of the image using Adobe Photoshop (CC 2017) software.

#### 2.3.7. Statistical Analysis

All experiments were performed in triplicate. The average and standard deviation of the particle size, zeta potential, and encapsulation efficiency results were calculated with the Microsoft Excel (2016) program.

## 3. Results and Discussion

### 3.1. Formation Principle of the Nanocapsules

The mixed solution of TB-BCG-BCP and EC (positive solvent) and the aqueous PVA solution (antisolvent) are injected at the same flow rate into both inlets of the CIJ mixer. The positive and antisolvent collided and mixed in a confined and narrow space within milliseconds. With the rapid increase in the proportion of antisolvent, the solubility of TB-BCG-BCP reached supersaturation and precipitated to form hydrophobic cores, while the hydrophilic group (-OH) of EC extended toward the aqueous phase and the hydrophobic group (C_2_H_5_O-) aggregated inward, thus encapsulating the hydrophobic core [18].

### 3.2. Microscopic Morphology and Structure

The morphology of the nanocapsules was examined using SEM and is shown in Figure 2. The images show that there is no adhesion between the capsules in the nanocapsules (Figure 2). Figure 2 shows that the overall shape of the nanocapsules was relatively regular and spherical, with a uniform size. The surface was smooth, without apparent cracks, holes, or folds.

### 3.3. FTIR Analysis of the Nanocapsules

The infrared spectra of pure TB-BCG-BCP, EC, EC/TB-BCG-BCP mixture, and nanocapsules loaded with TB-BCG-BCP are shown in Figure 3.

The infrared spectrum of EC (green line) reveals that the peak at 3481 cm^−1^ was the characteristic absorption peak of the -OH group. The double peak in the range of 2800–3000 cm^−1^ was due to the stretching vibration of the -CH_2_- group. The characteristic absorption peak at 1377 cm^−1^ was mainly due to the bending vibration of the -CH_3_ group. The peak at 1111 cm^−1^ might be due to the vibration of the -C-OH group. In the spectrum of pure TB-BCG-BCP (blue line), the characteristic absorption peak of the -OH group is at 3457 cm^−1^. The characteristic absorption peak of the benzene ring group of nanocapsules was noted at 1588 cm^−1^. However, the characteristic absorption peak of the -CH_3_ group was observed at 1451 cm^−1^, and the symmetric and asymmetric stretching vibrations of the -SO_2_- group were observed at 1161 and 1344 cm^−1^, respectively.

Compared to the infrared spectra of pure TB-BC-BCP and EC, the spectrum of the nanocapsules (black line) showed an enhancement of the -OH group at approximately 3400 cm^−1^. The characteristic absorption peaks of TB-BCG-BCP in the range of 1100–1600 cm^−1^ were significantly weakened. However, the characteristic absorption peak of TB-BCG-BCP was located in the spectrum of the mixture of EC and TB-BCG-BCP (red line). This demonstrates that TB-BCG-BCP was well encapsulated by EC.

### 3.4. Particle Size and Distribution

#### 3.4.1. Effect of the Flow Rate

The core-to-wall ratio (1:2) and the mixed solution concentration (6 mg/mL) were kept constant. The particle size, PDI, and size distribution of the nanocapsules at different flow rates are displayed in Figure 4. Figure 4 shows that as the flow rate increased from 20 to 40 mL/min, the particle size decreased from 181.47 ± 1.70 to 141.83 ± 0.98 nm and the size distribution became more uniform. However, when the flow rate was increased further (to 60 mL/min), the particle size became almost constant with increasing flow rate, and the size dispersion and PDI value also increased.

The flow rate is an important parameter during nanoparticle synthesis using a CIJ mixer. A higher flow rate results in a higher level of supersaturation, leading to a higher nucleation rate and facilitating the production of smaller and more homogeneous nanoparticles [12]. However, a flow rate that is too high can lead to low uniformity of the nanocapsules.

#### 3.4.2. Effect of the Core-to-Wall Ratio

The flow rate (40 mL/min) and the mixed solution concentration (6 mg/mL) were kept constant. The particle size, PDI, and size distribution of the nanocapsules at different core-to-wall mass ratios are shown in Figure 5. With a reduction in the core-to-wall ratio (1:1 to 1:2), the average particle size decreased from 174.13 ± 2.22 to 141.83 ± 0.98 nm, indicating a minimum value (Figure 5). When the core-to-wall ratio was 1:3, the particle size reached 160.20 ± 2.23 nm, which may be because the capsules absorbed more polymer during the growth phase. The variation in the PDI value and the size distribution show that the core-to-wall ratio has no significant effect on particle dispersity and uniformity.

#### 3.4.3. Effect of Mixed Solution Concentration

The flow rate (40 mL/min) and the core-to-wall ratio (1:2) were kept constant. Figure 6 shows the particle size, PDI, and size distribution of the nanocapsules at different concentrations of the mixed solution. At a concentration of 6 mg/mL, the average particle size of the nanocapsules was reduced to 141.83 ± 0.98 nm, and the PDI value was 0.125, indicating the smallest particle size and distribution at all concentrations measured (Figure 6). The average particle size and PDI value increased in small increments as the mixed solution concentration increased. It can be concluded that the particle size and PDI values increased as the mixed solution concentration increased from 6 to 10 mg/mL. In essence, the solution concentration was found to have a strong structure-directing effect in controlling the particle size for further drug uptake and delivery experimentation [19,20,21].

### 3.5. Dispersion Stability of Nanocapsules

The zeta potential test results of nanocapsule suspensions with different antisolvents are shown in Figure 7. In Figure 7, S1 represents nanocapsules with pure water as an antisolvent, while S2 represents nanocapsules with an antisolvent of aqueous PVA solution. Generally, the higher the absolute value of the zeta potential, the more mutually exclusive the nanocapsules will be, and thus, the more they can be firmly dispersed in the system. Conversely, the lower the absolute value of the zeta potential, the more the nanocapsules will tend to aggregate due to mutual attraction [22,23].

As shown in Figure 7, compared to before and after the addition of PVA to the antisolvent, the absolute value of the average zeta potential of the nanocapsules increased from 21.1 to 31.8 mV. This is due to the reduction in interfacial tension and stabilization of the steric hindrance caused by the appropriate amount of PVA, resulting in a steady state of the concentration, viscosity, and size of the dispersion system of the nanocapsules. The absolute value of the average zeta potential exceeded 30 mV, indicating good dispersion stability.

### 3.6. Encapsulation Efficiency (EE%)

The UV absorption spectra of different concentrations of TB-BCG-BCP-water standard solutions were examined and are shown in Figure 8a, and 433 nm was considered the maximum absorption value. Based on the absorbance of different concentrations of standard solutions at 433 nm, a linear fit was applied to obtain the standard curve, as shown in Figure 8. The linear fit represented the relationship between absorbance and TB-BCG-BCP concentration, and the standard equation was y = 0.0352x − 0.1998 (R^2^ = 0.99989).

The encapsulation efficiency was calculated using the method introduced in 2.3.5, and the results are shown in Figure 9a,b. Figure 9a shows that the maximum encapsulation efficiency was 75.20 ± 1.72% at a flow rate of 40 mL/min. At lower flow rates, it is difficult to supersaturate the core material to form a hydrophobic core. However, high flow rates can cause the newly formed nanocapsules’ structure to lose its integrity and breakdown under excessive mixing, resulting in partial leakage of the core material. It was concluded that both cases could lead to poor encapsulation rates.

As shown in Figure 9b, the encapsulation efficiency increased as the core-to-wall ratio decreased. When the core-to-wall ratio was high, the amount of EC was insufficient to encapsulate all the TB-BCG-BCP, and the core material agglomerated, which may lead to a lower encapsulation rate of the nanocapsules. As the EC ratio increased, more core material was encapsulated, and therefore, the encapsulation efficiency of the nanocapsules improved.

As shown in Figure 9c, the encapsulation efficiency decreased with increasing mixed solution concentration. This behavior can be attributed to the fact that TB-BCG-BCP and EC gradually precipitated out of the saturated system before they participated in the reaction, which may lead to waste of core and wall materials and a decrease in encapsulation efficiency.

### 3.7. Color Responsiveness of pH

For color responsiveness, 400 μL of the nanocapsule suspension was added dropwise to 150 μL of a buffer solution with an increasing pH (pH 3–10) (Figure 10). Figure 10 shows the significant change in nanocapsule color as the pH increased. The H-value range of the nanocapsules on the hue ring is shown in Figure 11. As the pH increased from 3 to 10, the H-value gradually increased from 47 to 241°, and the color shifted from yellow to green and then to blue. From pH 3–6, the H-value of the capsule was strongly influenced by the pH values and increased from 47 to 124°, and the color changed from yellow to yellow-green and then to dark green. From pH 7–10, the H-value was observed between 200 and 250°. Although the changes in pH have little impact on the H-value, they still showed a different blue color with a certain chromatic difference that is discernible to the naked eye.

## 4. Application of Sodium Alginate Hydrogel Fibers

Alginate hydrogel fibers are inherently flexible and can be shaped into irregular geometries, making them easier to use in various applications [24]. They are also typically transparent, which makes them suitable for developing smart detection products based on color changing. Therefore, we prioritized attempts to embed color-changing nanocapsules into sodium alginate (SA) hydrogel fibers (Figure 12 and Table 1).

We mixed the 3% w/v SA solution with the nanocapsule suspension in a 2:1 volume ratio and connected the syringe containing the mixture to an 18 G needle using a silicone tube. The syringe pump squeezed the syringe mixture into a 2% w/v CaCl_2_ coagulation bath at a rate of 10 mL/min. The hydrogel fibers were washed with deionized water and then collected.

The hydrogel fibers were divided into eight equal lengths and put into separate bottles with different buffer solutions in relation to pH (pH 3–10) (Figure 12 and Table 1). Figure 12 and Table 1 show the color changes and corresponding H-values of the hydrogels. The H-values indicated that the discoloration of the hydrogel fibers increased with increasing pH and was smaller compared to the H value. This situation can be attributed to the discoloration of the nanocapsules at the same pH but following the same trend. The pH color-changing nanocapsules can still achieve effective color changes in sodium alginate hydrogel fibers, which were expected to be used in medical dressings to detect the change in the pH of wound exudates.

## 5. Conclusions

In this study, wide-domain pH color-changing nanocapsules were prepared by the flash nanoprecipitation method with a new-type droplet-shaped confined impinging jet mixer. Mixing the three indicators at an optimum mixing ratio of 1:1:1, the pH color-changing domain of the mixed indicators extended to pH 3–10. The excellent preparation process for the nanocapsulation was as follows: the core-to-wall ratio was 1:2, the concentration of the mixed solution was 6 mg/mL, and the feed flow rate was 40 mL/min. Under these conditions, the nanocapsules obtained were as follows: the average particle size was 141.83 ± 0.98 nm, the PDI was 0.125 ± 0.01, the zeta potential was −31.83 ± 0.23 mV, and the encapsulation efficiency was 75.20 ± 1.72%, with good storage and dispersion stability. The nanocapsules exhibited a continuous color change from yellow to green and then to blue when the pH values changed from 3 to 10. In addition, the nanocapsules can be mixed directly with sodium alginate solutions to form hydrogels or hydrogel fibers by ionic cross-linking with CaCl_2_ solutions. The embedded loading of the nanocapsules and the transparency of the hydrogel allow for successful color-changing sensing under acidic and alkaline conditions, which shows the potential application in the field of medical dressings, paper diapers, and sports straps to monitor the pH values of wound exudate, urine, and sweat, respectively.

## Figures and Tables

**Figure 1 materials-15-08787-f001:**
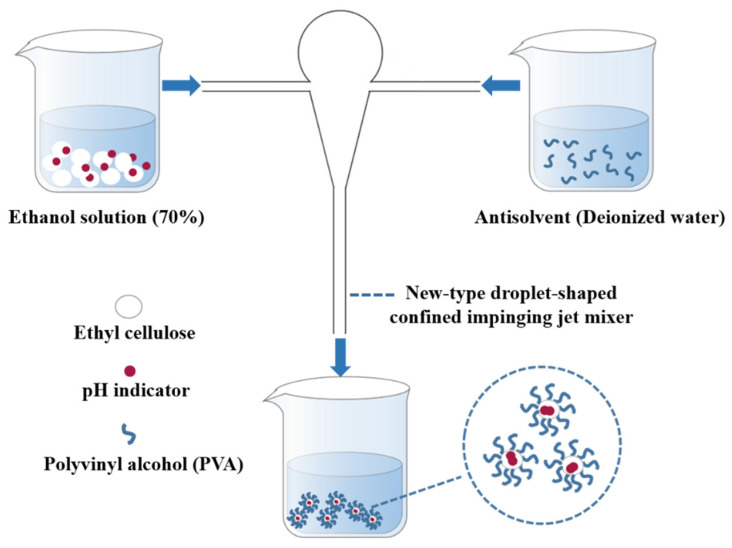
Schematic diagram showing the preparation of nanocapsules.

**Figure 2 materials-15-08787-f002:**
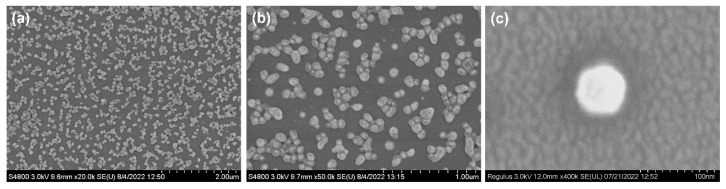
SEM images of nanocapsules examined at different configurations: (**a**) 20,000×, (**b**) 50,000×, and (**c**) 400,000×.

**Figure 3 materials-15-08787-f003:**
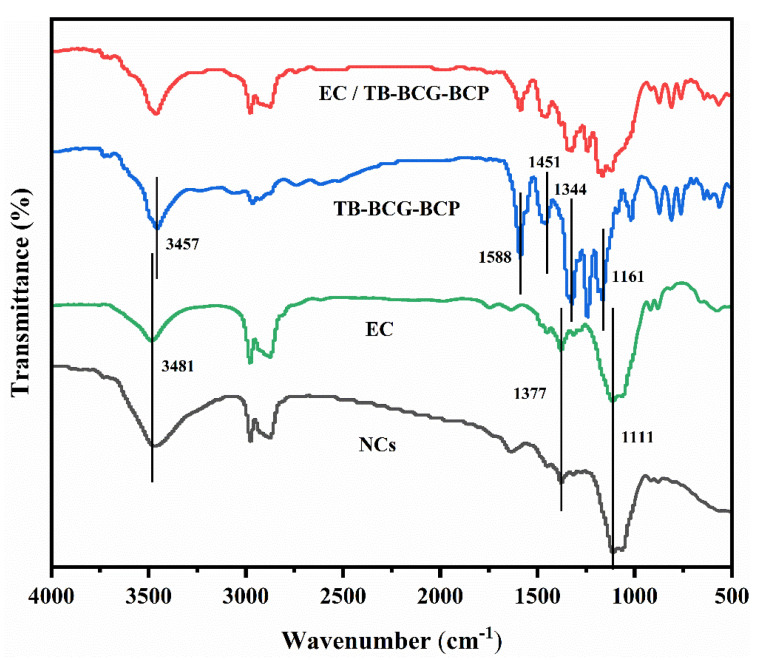
FTIR spectra of pure TB-BCG-BCP, EC, an EC/TB-BCG-BCP mixture, and nanocapsules loaded with TB-BCG-BCP.

**Figure 4 materials-15-08787-f004:**
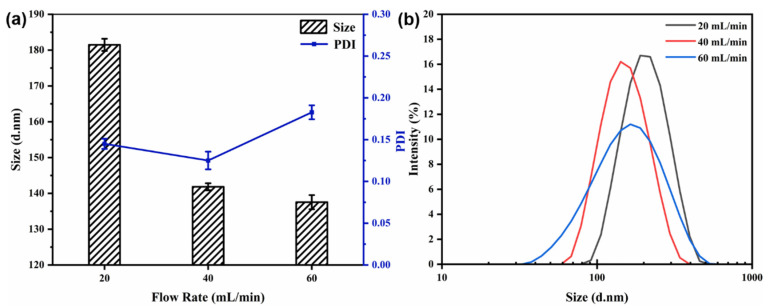
Effect of flow rate on particle size and PDI (**a**) and size distribution (**b**) of nanocapsules.

**Figure 5 materials-15-08787-f005:**
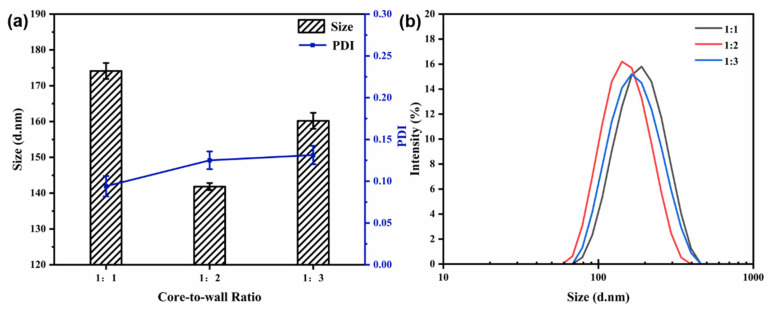
Effect of core-to-wall ratio on particle size and PDI (**a**) and size distribution (**b**) of nanocapsules.

**Figure 6 materials-15-08787-f006:**
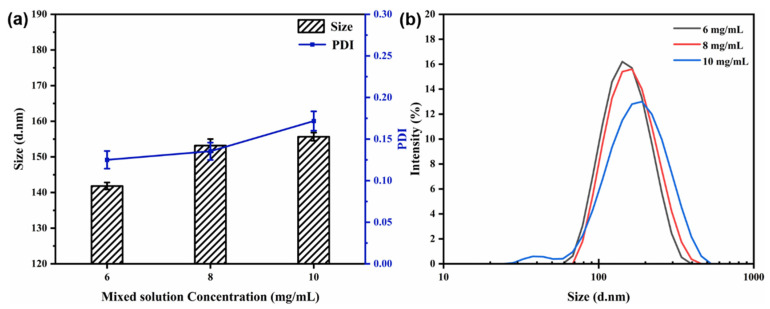
Effect of mixed solution concentration on particle size and PDI (**a**) and size distribution (**b**) of nanocapsules.

**Figure 7 materials-15-08787-f007:**
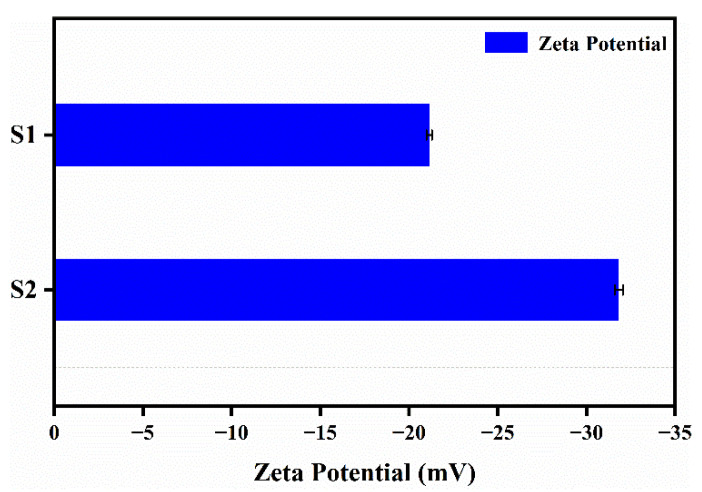
Zeta potential of nanocapsules (S1 and S2) with pure water as an antisolvent for S1 and an antisolvent of an aqueous PVA solution for S2.

**Figure 8 materials-15-08787-f008:**
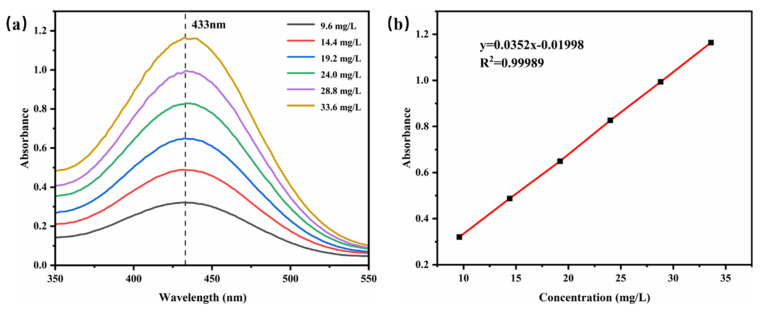
UV absorption spectra (**a**) and standard curve (**b**) of the TB-BCG-BCP-water standard solution.

**Figure 9 materials-15-08787-f009:**
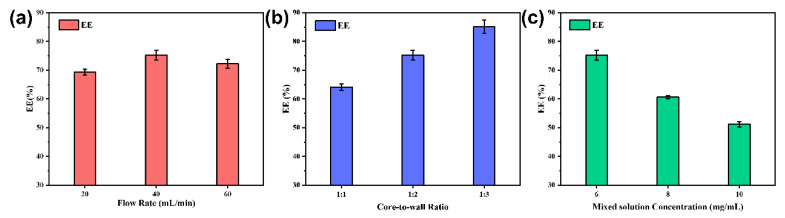
Encapsulation efficiency of nanocapsules at different flow rates (**a**) at different core-to-wall ratios (**b**) at different concentrations of mixed solutions (**c**).

**Figure 10 materials-15-08787-f010:**
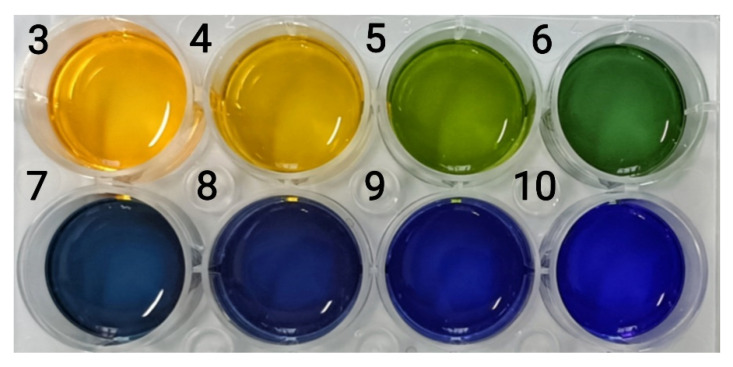
Color of the nanocapsule suspension at pH 3–10 (the numbers represent pH values).

**Figure 11 materials-15-08787-f011:**
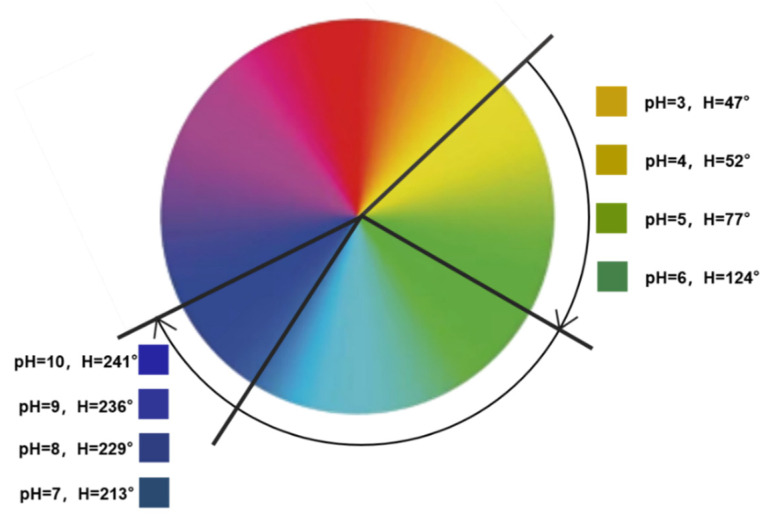
The H-value range of nanocapsules under different pH values (pH 3–10).

**Figure 12 materials-15-08787-f012:**
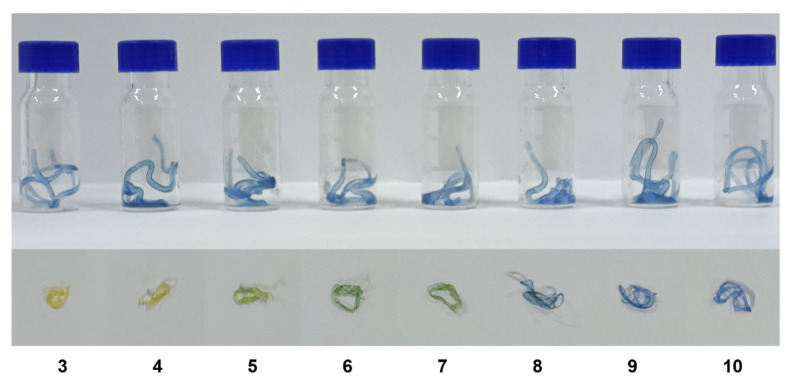
Discoloration of hydrogel fibers at different pH values (pH 3–10) (the numbers represent pH values).

**Table 1 materials-15-08787-t001:** H-values corresponding to discoloration of hydrogel fibers at different pH ranges (pH 3–10).

pH	3	4	5	6	7	8	9	10
Colors displayed								
H-value (°)	54.0	61.6	67.8	92.5	95.4	207.1	213.7	218.1

## Data Availability

Not applicable.

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
