# Peer review of "Preparation of Wide-Domain pH Color-Changing Nanocapsules and Application in Hydrogel Fibers"

_materials, 2022, doi:10.3390/ma15248787_

Round 1
Reviewer 1 Report
The author Xuemei Hou has reported manuscript with title Preparation of wide-domain pH colour-changing nanocapsules and application in hydrogel fibers. The article is well written and has very good results with different style presentation. I recommended it for publication after minor changes.
Following are main key points.
1. The instroduction is very short, There is need to write more about importance of hydrogel fiber. nanocapsules etc.
2. In method section, there is need to write model of instruments like SEM, FTIIR.
3. Re-arrange Figure 2 with little distance between them.
4. There is lack of discussion. please update reference with discussion
5. Particle size, PDI value, flow rate, please see references.
6. figure 9 should rearrage into one figure.
Author Response
We thank you for your valuable comments and suggestions. Your comments have been effective in improving the manuscript. We will respond to your comments point by point below.
Point 1: The introduction is very short, There is need to write more about importance of hydrogel fiber. nanocapsules etc.
Response 1: The significance and importance of nanocapsules and hydrogel fibers have been added to the introduction of the original article, which has been highlighted in yellow.
Point 2: In method section, there is need to write model of instruments like SEM, FTIIR.
Response 2: All model of instruments has been added to the original article. Marking has been done in the original article.
Point 3: Re-arrange Figure 2 with little distance between them.
Response 3: Figure 2 has been rearranged and modified in the original article.
Point 4: There is lack of discussion. please update reference with discussion.
Point 5: Particle size, PDI value, flow rate, please see references.
DOI: 10.1021/acsomega.9b00119
DOI: 10.1016/j.mtchem.2018.02.003
DOI: 10.1039/C8NA00343B
Response 4&5: This part of the discussion has been added, and the references you suggested have been cited. I have marked them in the original article.
Point 6: figure 9 should rearrange into one figure.
Response 6: Figure 9 has been rearranged and modified in the original article.

Reviewer 2 Report
The results obtained in this research will be applied in the various fields.
But a few explanations for the following things must be necessary.
[1] Do the dye species encapsulated in the nano-capsules or in the hydrogel
fibers leak out naturally in the water phase?
[2] There is no explanation for BCP in 2.1 Materials.
[3] Is there the mutual interference among the dye species encapsulated for
the pH color changing?
Author Response
We thank you for your valuable comments and suggestions. Your comments have been effective in improving the manuscript. We will respond to your comments point by point below.
Point 1: Do the dye species encapsulated in the nano-capsules or in the hydrogel fibers leak out naturally in the water phase?
Response 1: The nanocapsules or hydrogel fibers in the nanocapsules are immersed in a solution environment when the core material diffuses, resulting in some degree of leakage, which is common to microcapsules in terms of slow release. Therefore, as the reviewer has seen, the experimental immersion method into pH-buffered solutions used in this paper may result in a certain degree of leakage. However, in practical applications, the amount of liquid contacted will be so small that leakage will not occur.
Point 2: There is no explanation for BCP in 2.1 Materials.
Response 2: The explanation of the BCP has been added to the 2.1 Materials.
Marking has been done in the original article.
Point 3: Is there the mutual interference among the dye species encapsulated for the pH color changing?
Response 3: The three indicators do not interfere with each other when mixed, and mixing in the right proportions will instead increase the colour change domain, as shown in Figures 10 and 11 in Section 3.7 of the text, where the nanocapsule suspensions show different colours and H values at pH 3-10. Rukchon's research also showed that the smart label used to determine the freshness of chicken was prepared by mixing two indicators, bromocresol blue and methyl red, which expanded the colour change field (Rukchon, C., Nopwinyuwong, A., Trevanich, S., Jinkarn, T., & Suppakul, P. (2014). Development of a food spoilage indicator for monitoring freshness of skinless chicken breast. Talanta, 130, 547-554.).

Reviewer 3 Report
In this paper, the authors achieved the formation of nanoparticle composed of pH colour-changing dye and ethyl cellulose with the flash nanoprecipitation method using the confined impinging jets mixer. The authors checked the effect of flow rate, core-to-wall ratio, and concentration of compositions on the nanoparticle properties such as size, PDI and encapsulation efficiency, and then they concluded that flow rate of 40 mL/min, core-to-wall ratio of 1:2, and concentration of 6 mg/mL are best. The dye-loaded nanoparticle showed pH-responsive color change with the pH range of 3-10. This looks an interesting work but the current manuscript has many points to be revised before publication.
1.
First of all, in scientific reports, the authors have to use past tense for results and present tense for discussion and opinion. The current manuscript cannot be permitted to be published because all sentences are written in the present tense and the reader cannot distinguish correctly the scientific results and authors’ thinking.
2.
Is this concept of nanoparticle formation with polymer and pH-sensitive dye new? The authors should explain the background and introduce the similar previous report about the developments of pH color-changing materials, pH color-changing gel, or domain width of rivals’ works. The current introduction explained the significance of pH color-changing material and the principle of FNP method but did not mentioned about the position of this paper in the pH color-changing material field.
3,
The pH-color changing materials of TB-BCG-BCP with mixing ratio of 1:1:1 is new? What was the reason to choose their probes and the ratio? If there are previous reports, the authors should cite them. If the mixture of TB-BCG-BCP is new, the authors should focus on and discuss the pH-responsive color change and H-value more carefully.
4.
In the discussion about the effect of flow rate, the authors explained that higher flow rate will produce a higher nucleation rate and so on… Have the contents of this paragraph been reported? Or just authors’ hypothesis? If former, please cite literatures. If latter, please modify with “hypothesized” or “expected” or any other words to make it easy to understand that is not fact.
5.
I did not find the condition to prepare samples of Figure 4, 5 and 6. In Figure 4, the authors changed the flow rate but what were the core-to-ratio and concentration? Figure 5 and 6, too. Please add them in materials and methods.
6.
Dispersion stability of nanoparticle was evaluated by zeta-potential. However, zeta-potential means surface charge of nanoparticle first. Higher positive or negative charge induce electrostatic repulsion among nanoparticle, resulting in the dispersion stability. The authors should discuss the reason of highly negative charge of nanoparticle by using PVA and should not regard the negative charge as a result of higher dispersion stability.
7.
Authors estimated EE in Figure 9. How many samples measured? How was error bar estimated? How about the P value?
8.
The nanoparticle showed higher and lower sensitivity (difference of H-value) against the pH range of 3-6 and 7-10. What was its reason? The authors should discuss it.
9.
In the section of “Application in sodium alginate hydrogel fibers”, the authors should explain the purpose in the first sentence.
10.
In the section 4. “SA solution of 3% w/v was mixed with nanocapsule suspension at a volume ratio of 2:1 under optimized conditions.” Who optimized the condition? If it was the authors, please explain the optimization. If it was done by another teams, please cite the literature.
11.
H-value of nanoparticle at pH 7 was 213 but that of nanoparticle-suspended gel at pH 7 was 95. Please discuss the difference. Also, all pH conditions, their values are slightly different between nanoparticle and nanoparticle-suspended gel. What was the reason? This is very important point because, I think, gel property (thickness, cross-linking ratio, or polymer kinds) affects pH sensitivity.
Minor
3.2. As shown in figure 2 > As shown in Figure 2
Flow rate is an important parameter when preparing nanoparticles using CIJM. > Flow rate is an important parameter when preparing nanoparticles using CIJ mixer.
Authors used Figure X and Fig.X randomly in text but they should be unified.

Author Response
We thank you for your valuable comments and suggestions. Your comments have been effective in improving the manuscript. We will respond to your comments point by point below.
Point 1: First of all, in scientific reports, the authors have to use past tense for results and present tense for discussion and opinion. The current manuscript cannot be permitted to be published because all sentences are written in the present tense and the reader cannot distinguish correctly the scientific results and authors’ thinking.
Response 1: Tense changes and English expression changes have been made throughout the text.
Point 2: Is this concept of nanoparticle formation with polymer and pH-sensitive dye new? The authors should explain the background and introduce the similar previous report about the developments of pH color-changing materials, pH color-changing gel, or domain width of rivals’ works. The current introduction explained the significance of pH color-changing material and the principle of FNP method but did not mentioned about the position of this paper in the pH color-changing material field.
Response 2: Relevant studies have been added in the INTRODUCTION section of the original article. Marking has been done in the original article.
Point 3: The pH-color changing materials of TB-BCG-BCP with mixing ratio of 1:1:1 is new? What was the reason to choose their probes and the ratio? If there are previous reports, the authors should cite them. If the mixture of TB-BCG-BCP is new, the authors should focus on and discuss the pH-responsive color change and H-value more carefully.
Response 3: pH colour-changing chemical indicators generally have a narrow colour change range, and compounding multiple indicators is a viable method to extend the indicator's colour change range. For example, Rukchon mixed two indicators, bromocresol blue and methyl red, and prepared a smart label for monitoring the freshness of chicken, where the mixed indicator broadened the field of colour change (Rukchon, C., Nopwinyuwong, A., Trevanich, S., Jinkarn, T., & Suppakul, P. (2014). Development of a food spoilage indicator for monitoring freshness of skinless chicken breast. Talanta, 130, 547-554.). The three indicators Thymol Blue (TB), Bromocresol Green (BCG) and Bromocresol Purple (BCP) were selected because their physicochemical properties are consistent with the experimental principles of this study, and their respective colour change domains do not interfere with each other when mixed. Moreover, we have found that the colour change range is more expansive when the mixture ratio is 1:1:1 after extensive experiments.
Point 4: In the discussion about the effect of flow rate, the authors explained that higher flow rate will produce a higher nucleation rate and so on… Have the contents of this paragraph been reported? Or just authors’ hypothesis? If former, please cite literatures. If latter, please modify with “hypothesized” or “expected” or any other words to make it easy to understand that is not fact.
Response 4: References to this section have been included in the original article.
Point 5: I did not find the condition to prepare samples of Figure 4, 5 and 6. In Figure 4, the authors changed the flow rate but what were the core-to-ratio and concentration? Figure 5 and 6, too. Please add them in materials and methods.
Response 5: Missing conditions have been added and marked in the original article. Marking has been done in the original article.
Point 6: Dispersion stability of nanoparticle was evaluated by zeta-potential. However, zeta-potential means surface charge of nanoparticle first. Higher positive or negative charge induce electrostatic repulsion among nanoparticle, resulting in the dispersion stability. The authors should discuss the reason of highly negative charge of nanoparticle by using PVA and should not regard the negative charge as a result of higher dispersion stability.
Response 6: After the adsorption of PVA on the particle surface, a spatial site resistance is generated, which causes the shear surface to move in a direction away from the particle surface, increasing the repulsive force and dispersion stability. At this point, the zeta potential shows an increase in absolute value, which results in a high negative charge. In agreement with your view, the higher positive or negative charge causes electrostatic repulsion between the nanoparticles, resulting in dispersion stability. The apparent increase in the absolute value of the zeta potential of the nanocapsules when using PVA is evidence in my judgement of the increased dispersion stability of the nanocapsules.
Point 7: Authors estimated EE in Figure 9. How many samples measured? How was error bar estimated?
Response 7: A total of three factors were explored in the process of measuring EE on EE. Each factor contains three levels, each of which measures five samples. The error bar is a line segment drawn in the direction indicating the magnitude of the measured value, using the arithmetic mean of the measured value as the midpoint, with half the length of the line segment equal to the (standard or extended) uncertainty. The error bars are derived from the origin statistics.
Point 8: The nanoparticle showed higher and lower sensitivity (difference of H-value) against the pH range of 3-6 and 7-10. What was its reason? The authors should discuss it.
Response 8: The mixed indicator achieves a wide range of colour changes, but there are differences in the sensitivity of the colour change in different pH. This will motivate us to further develop more sensitive wide range colour change indicators. Also, certain electrolytes contained in the titration solution have the property of absorbing light waves of different wavelengths, which can affect the colour depth, hue and colour change sensitivity of the nanocapsules. However, the different colour changes can still be distinguished by accurate colour recognition software, compensating for the lack of observation of colour changes through the naked eye.
Point 9: In the section of “Application in sodium alginate hydrogel fibers”, the authors should explain the purpose in the first sentence.
Response 9: We have explained the purpose by adding the first paragraph of the section "Application in sodium alginate hydrogel fibers" and have marked it in the original.
Point 10: In the section 4. “SA solution of 3% w/v was mixed with nanocapsule suspension at a volume ratio of 2:1 under optimized conditions.”Who optimized the condition? If it was the authors, please explain the optimization. If it was done by another teams, please cite the literature.
Response 10: We have tuned the different parameters after several experiments while referring to the optimisation conditions in Tamayol's study on flexible pH-sensing hydrogel fibres for epidermal applications (Tamayol, A., Akbari, M., Zilberman, Y., Comotto, M., Lesha, E., Serex, L., ... & Khademhosseini, A. (2016). Flexible pH‐sensing hydrogel fibers for epidermal applications. Advanced healthcare materials, 5(6), 711-719.). The process parameters for the hydrogel fibres loaded with pH-changing nanocapsules prepared as described in section 4 were determined. Under this optimised condition, the hydrogels were well formed and had good colour-change properties.
Point 11: H-value of nanoparticle at pH 7 was 213 but that of nanoparticle-suspended gel at pH 7 was 95. Please discuss the difference. Also, all pH conditions, their values are slightly different between nanoparticle and nanoparticle-suspended gel. What was the reason? This is very important point because, I think, gel property (thickness, cross-linking ratio, or polymer kinds) affects pH sensitivity.
Response 11: The difference in the amount of colour-changing nanocapsules contained in the nanocapsule suspension and the nanocapsule-loaded hydrogel is the main reason for the difference in H-values at the same pH. At pH = 7, the difference in H-values was greater than at other pH values, probably due to the performance of the hybrid indicator. As explained in question 8, the hybrid indicator achieves a wide range of colour changes, but at this stage, it is not possible to achieve a large colour difference for every change in pH. The authors acknowledge the reviewer's view that differences such as the degree of cross-linking of the hydrogel, the thickness of the fibres and even the type of polymer used in the hydrogel can affect pH sensitivity and colour change performance. The influence of these factors on colour change sensitivity will be looked at and discussed in our subsequent studies.
